# Projected Stein Variational Gradient Descent

**Peng Chen**      **Omar Ghattas**
Oden Institute for Computational Engineering and Sciences
The University of Texas at Austin
Austin, TX 78712.
`{peng, omar}@oden.utexas.edu`

## Abstract

The curse of dimensionality is a longstanding challenge in Bayesian inference in high dimensions. In this work, we propose a projected Stein variational gradient descent (pSVGD) method to overcome this challenge by exploiting the fundamental property of intrinsic low dimensionality of the data informed subspace stemming from ill-posedness of such problems. We adaptively construct the subspace using a gradient information matrix of the log-likelihood, and apply pSVGD to the much lower-dimensional coefficients of the parameter projection. The method is demonstrated to be more accurate and efficient than SVGD. It is also shown to be more scalable with respect to the number of parameters, samples, data points, and processor cores via experiments with parameters dimensions ranging from the hundreds to the tens of thousands.

## 1   Introduction

Given observation data for a system with unknown parameters, Bayesian inference provides an optimal probability framework for learning the parameters by updating their prior distribution to a posterior distribution. However, many conventional methods for solving high-dimensional Bayesian inference problems face the curse of dimensionality, i.e., the computational complexity grows rapidly, often exponentially, with respect to (w.r.t.) the number of parameters. To address the curse of dimensionality, the intrinsic properties of the posterior distribution, such as its smoothness, sparsity, and intrinsic low-dimensionality, have been exploited to reduce the parameter correlation and develop efficient methods whose complexity grows slowly or remains the same with increasing dimension. By exploiting the geometry of the log-likelihood function, accelerated Markov chain Monte Carlo (MCMC) methods have been developed to reduce the sample correlation or increase effective sample size independent of the dimension [21, 25, 27, 17, 18, 2]. Nevertheless, these random and essentially serial sampling methods remain prohibitive for large-scale inference problems with expensive likelihoods. Deterministic methods using sparse quadratures [32, 30, 10, 11] were shown to converge rapidly with dimension-independent rates for problems with smooth and sparse posteriors. However, for posteriors lacking smoothness or sparsity, the convergence deteriorates significantly, despite incorporation of Hessian-based transformations [31, 12].

Transport-based variational inference is another type of deterministic method that seeks a transport map in a function space (represented by, e.g., polynomials, kernels, or neural networks) that pushes the prior to the posterior by minimizing the difference between the transported prior and the posterior, measured in, e.g., Kullback–Leibler divergence [26, 24, 4, 3, 20]. In particular, kernel-based Stein variational methods, using gradient-based (SVGD) [24, 16, 23, 8] and Hessian-based (SVN) [19, 35] optimization methods, are shown to achieve fast convergence in relatively low dimensions. Nonetheless, the convergence and accuracy of these methods deteriorates in high dimensions due to the curse of dimensionality in kernel representation. This can be partially addressed by a localized

SVGD on Markov blankets, which relies on conditional independence of the target distribution [37, 34], or by parameter projection for dimension reduction in pSVN [14] and lazy maps [3].

**Contributions**: Here, we propose, analyze, and apply a projected SVGD method to tackle the curse of dimensionality for high-dimensional nonlinear Bayesian inference problems, which relies on the fundamental property that the posterior effectively differs from the prior only in a low-dimensional subspace of high-dimensional parameters, see [6, 33, 22, 18, 12, 13, 7, 3, 14] and references therein. Specifically, our contributions are: (1) we propose dimension reduction by projecting the parameters to a low-dimensional subspace constructed using the gradient of the log-likelihood, and push the prior samples of the projection coefficients to their posterior by pSVGD; (2) we prove the equivalence of the projected transport in the coefficient space and the transport in the projected parameter space; (3) we propose adaptive and parallel algorithms to efficiently approximate the optimal profile function and the gradient information matrix for the construction of the subspace; and (4) we demonstrate the accuracy (compared to SVGD) and scalability of pSVGD w.r.t. the number of parameters, samples, data points, and processor cores by classical high-dimensional Bayesian inference problems.

The major differences of this work compared to pSVN [14] are: (1) pSVGD uses only gradient information of the log-likelihood, which is available for many models, while pSVN requires Hessian information, which is challenging for complex models and codes in practical applications; (2) the upper bound for the pSVGD projection error w.r.t. the posterior, based on a result from [36], is sharper than that for pSVN; (3) we prove the equivalence of the projected transport for the coefficient and the transport for the projected parameters; (4) we also test new benchmark examples and investigate the convergence of pSVGD w.r.t. the number of parameters and the scalability of pSVGD w.r.t. the number of data points.

## 2 Preliminaries

Let $x \in \mathbb{R}^d$ denote a random parameter of dimension $d \in \mathbb{N}$, which has a continuous prior density $p_0 : \mathbb{R}^d \to \mathbb{R}$. Let $y = \{y_i\}_{i=1}^s$ denote a set of i.i.d. observation data. Let $f(x) := \prod_{i=1}^s p(y_i|x)$ denote, up to a multiplicative constant, a continuous likelihood of $y$ at given $x$. Then the posterior density of parameter $x$ conditioned on data $y$, denoted as $p(\cdot) : \mathbb{R}^d \to \mathbb{R}$, is given by Bayes' rule as

$$p(x) = \frac{1}{Z}f(x)p_0(x), \tag{1}$$

where $Z$ is the normalization constant defined as

$$Z = \int_{\mathbb{R}^d} f(x)p_0(x)dx, \tag{2}$$

whose computation is typically intractable, especially for a large $d$. The central task of Bayesian inference is to draw samples of parameter $x$ from its posterior with density $p$, and compute some statistical quantity of interest, e.g., the mean and variance of the parameter $x$ or some function of $x$.

SVGD is one type of variational inference method that seeks an approximation of the posterior density $p$ by a function $q^*$ in a predefined function set $\mathcal{Q}$, which is realized by minimizing the Kullback–Leibler (KL) divergence that measures the difference between two densities, i.e.,

$$q^* = \arg\min_{q \in \mathcal{Q}} D_{\text{KL}}(q|p), \tag{3}$$

where $D_{\text{KL}}(q|p) = \mathbb{E}_{x \sim q}[\log(q/p)]$, i.e., the average of $\log(q/p)$ with respect to the density $q$, which vanishes when $q = p$. In particular, a transport based function set is considered as $\mathcal{Q} = \{T_\sharp p_0 : T \in \mathcal{T}\}$, where $T_\sharp$ is a pushforward map that pushes the prior density to a new density $q := T_\sharp p_0$ through a transport map $T(\cdot) : \mathbb{R}^d \to \mathbb{R}^d$ in a space $\mathcal{T}$. Let $T$ be given by

$$T(x) = x + \epsilon\phi(x), \tag{4}$$

where $\phi : \mathbb{R}^d \to \mathbb{R}^d$ is a differentiable perturbation map w.r.t. $x$, and $\epsilon > 0$ is small enough so that $T$ is invertible. It is shown in [24] that

$$\nabla_\epsilon D_{\text{KL}}(T_\sharp p_0 | p)\big|_{\epsilon=0} = -\mathbb{E}_{x \sim p_0}[\text{trace}(\mathcal{A}_p\phi(x))], \tag{5}$$

where $\mathcal{A}_p$ is the Stein operator given by

$$\mathcal{A}_p\phi(x) = \nabla_x \log p(x)\phi(x)^T + \nabla_x\phi(x). \tag{6}$$

Based on this result, a practical SVGD algorithm was developed in [24] by choosing the space $\mathcal{T} = (\mathcal{H}_d)^d = \mathcal{H}_d \times \cdots \times \mathcal{H}_d$, a tensor product of a reproducing kernel Hilbert space (RKHS) $\mathcal{H}_d$ with kernel $k(\cdot, \cdot) : \mathbb{R}^d \times \mathbb{R}^d \to \mathbb{R}$. SVGD updates samples $x_1^0, \ldots, x_N^0$ drawn from the prior $p_0$ as

$$x_m^{\ell+1} = x_m^\ell + \epsilon_l \hat{\phi}_\ell^*(x_m^\ell), \quad m = 1, \ldots, N, \ell = 0, 1, \ldots \tag{7}$$

where $\epsilon_l$ is a step size or learning rate, and $\hat{\phi}_\ell^*(x_m^\ell)$ is an approximate steepest direction given by

$$\hat{\phi}_\ell^*(x_m^\ell) = \frac{1}{N} \sum_{n=1}^N \nabla_{x_n^\ell} \log p(x_n^\ell) k(x_n^\ell, x_m^\ell) + \nabla_{x_n^\ell} k(x_n^\ell, x_m^\ell). \tag{8}$$

The kernel $k$ plays a critical role in pushing the samples to the posterior. One choice is Gaussian

$$k(x, x') = \exp\left(-\frac{||x - x'||_2^2}{h}\right), \tag{9}$$

where $h$ is the bandwith, e.g., $h = \text{med}^2 / \log(N)$ with med representing the median of sample distances [24]. However, it is known that the kernel suffers from the *curse of dimensionality* for large $d$ [28, 37, 34], i.e., the kernel is degenerate and becomes close to delta function for large $d$, which leads to samples not representative of the posterior, as observed in [37, 34], and also demonstrated in Section 4 by the inaccurate posterior sample variance and large testing error.

## 3 Projected Stein Variational Gradient Descent

To tackle the curse of dimensionality of SVGD, we exploit one fundamental property of many high-dimensional Bayesian inference problems — the posterior only effectively differs from the prior in a relatively low-dimensional subspace due to the ill-posedness or over-parametrization of the inference problems, see many examples and some proofs in, e.g., [1, 5, 6, 33, 22, 18, 12, 36, 13, 7, 3, 14]. In the following, we present the method to find the subspace and project the parameter to the subspace for dimension reduction in Section 3.1, which leads to the development of pSVGD in Section 3.2. Then we present practical algorithms of pSVGD and adaptive pSVGD in Section 3.3.

### 3.1 Dimension reduction by projection

We find the subspace in which the posterior evidently differs from the prior using a gradient-based parameter sensitivity of the log-likelihood function [36]. More specifically, by $H \in \mathbb{R}^{d \times d}$ we denote a gradient information matrix, which is defined as the average of the outer product of the gradient of the log-likelihood w.r.t. the posterior, i.e.,

$$H = \int_{\mathbb{R}^d} (\nabla_x \log f(x))(\nabla_x \log f(x))^T p(x) dx. \tag{10}$$

By $(\lambda_i, \psi_i)_{i=1}^r$ we denote the dominant eigenpairs of $(H, \Gamma)$, with $\Gamma$ representing the covariance of the parameter $x$ w.r.t. its prior, i.e., $(\lambda_i, \psi_i)_{i=1}^r$ correspond to the $r$ largest eigenvalues $\lambda_1 \geq \cdots \geq \lambda_r$,

$$H\psi_i = \lambda_i \Gamma \psi_i. \tag{11}$$

Given $H$, which is practically computed in Section 3.3, the generalized eigenvalue problem (11) can be efficiently solved by a randomized algorithm [29] that only requires $O(r)$ matrix vector product. We make the following key observation: *The eigenvalue $\lambda_i$ measures the sensitivity of the data w.r.t. the parameters along direction $\psi_i$, i.e., the data mostly inform parameters in directions $\psi_i$ corresponding to large eigenvalues $\lambda_i$. For $i$ with small $\lambda_i$, close to zero, the variation of the likelihood $f$ in direction $\psi_i$ is negligible, so the posterior is close to the prior in direction $\psi_i$.*

We define a linear projector of rank $r$, $P_r : \mathbb{R}^d \to \mathbb{R}^d$, as

$$P_r x := \sum_{i=1}^r \psi_i \psi_i^T x = \Psi_r w, \quad \forall x \in \mathbb{R}^d, \tag{12}$$

where $\Psi_r := (\psi_1, \ldots, \psi_r) \in \mathbb{R}^{d \times r}$ represents the projection matrix and $w := (w_1, \ldots, w_r)^T \in \mathbb{R}^r$ is the coefficient vector with element $w_i := \psi_i^T x$ for $i = 1, \ldots, r$. For this projection, we seek a

*profile function* $g : \mathbb{R}^d \to \mathbb{R}$ such that $g(P_r x)$ is a good approximation of the likelihood function $f(x)$. For a given profile function, we define a projected density $p_r : \mathbb{R}^d \to \mathbb{R}$ as

$$p_r(x) := \frac{1}{Z_r} g(P_r x) p_0(x), \tag{13}$$

where $Z_r := \mathbb{E}_{x \sim p_0}[g(P_r x)]$. It is shown in [36] that an optimal profile function $g^*$ exists such that

$$D_{\mathrm{KL}}(p|p_r^*) \leq D_{\mathrm{KL}}(p|p_r), \tag{14}$$

where $p_r^*$ is defined as in (13) with an optimal profile function $g^*$. Moreover, under certain mild assumptions for the prior (sub-Gaussian) and the likelihood function (whose gradient has the second moment w.r.t. the prior), the upper bound is shown (sharper than that for pSVN in [14]) as

$$D_{\mathrm{KL}}(p|p_r^*) \leq \frac{\gamma}{2} \sum_{i=r+1}^{d} \lambda_i, \tag{15}$$

for a constant $\gamma > 0$ independent of $r$, which implies small projection error for the posterior when $\lambda_i$ decay fast. The optimal profile function $g^*$ is nothing but the marginal likelihood given by

$$g^*(P_r x) = \int_{X_\perp} f(P_r x + \xi) p_0^\perp(\xi | P_r x) d\xi, \tag{16}$$

where $X_\perp$ is the complement of the subspace $X_r$ spanned by $\psi_1, \ldots, \psi_r$, and

$$p_0^\perp(\xi | P_r x) = p_0(P_r x + \xi) / p_0^r(P_r x) \text{ with } p_0^r(P_r x) = \int_{X_\perp} p_0(P_r x + \xi) d\xi. \tag{17}$$

We defer a practical computation of the optimal profile function to Section 3.3.

## 3.2  Projected Stein Variational Gradient Descent

By the projection (12), we consider a decomposition of the prior for the parameter $x = x^r + x^\perp$ as

$$p_0(x) = p_0^r(x^r) p_0^\perp(x^\perp | x^r), \tag{18}$$

where the marginals $p_0^r$ and $p_0^\perp$ are defined in (17). Moreover, since $p_0^r$ only depends on $x^r = P_r x = \Psi_r w$, we define a prior density for $w$ as

$$\pi_0(w) = p_0^r(\Psi_r w). \tag{19}$$

Then we define a joint (posterior) density for $w$ at the optimal profile function $g = g^*$ in (13) as

$$\pi(w) = \frac{1}{Z_w} g(\Psi_r w) \pi_0(w), \tag{20}$$

where the normalization constant $Z_w = E_{w \sim \pi_0}[g(\Psi_r w)]$. It is easy to see that $Z_w = Z_r$, where $Z_r$ is in (13), and the projected density in (13) can be written as

$$p_r(x) = \pi(w) p_0^\perp(x^\perp | \Psi_r w). \tag{21}$$

Therefore, to sample $x$ from $p_r$, we only need to sample $w$ from $\pi$ and $x^\perp$ from $p_0^\perp(x^\perp | \Psi_r w)$.

To sample from the posterior $\pi$ in (20), we employ the SVGD method presented in Section 2 in the coefficient space $\mathbb{R}^r$, with $r < d$. Specifically, we define a projected transport map $T^r : \mathbb{R}^r \to \mathbb{R}^r$ as

$$T^r(w) = w + \epsilon \phi^r(w), \tag{22}$$

with a differentiable perturbation map $\phi^r : \mathbb{R}^r \to \mathbb{R}^r$, and a small enough $\epsilon > 0$ such that $T^r$ is invertible. Following the argument in [24] on the result (5) for SVGD, we obtain

$$\nabla_\epsilon D_{\mathrm{KL}}(T^r_\sharp \pi_0 | \pi)\big|_{\epsilon=0} = -\mathbb{E}_{w \sim \pi_0}[\mathrm{trace}(\mathcal{A}_\pi \phi^r(w))], \tag{23}$$

where $\mathcal{A}_\pi$ is the Stein operator for $\pi$ given by

$$\mathcal{A}_\pi \phi^r(w) = \nabla_w \log \pi(w) \phi^r(w)^T + \nabla_w \phi^r(w). \tag{24}$$

Using a tensor product of RKHS $\mathcal{H}_r$ with kernel $k^r(\cdot, \cdot) : \mathbb{R}^r \times \mathbb{R}^r \to \mathbb{R}$ for the approximation of $\phi^r \in (\mathcal{H}_r)^r = \mathcal{H}_r \times \cdots \times \mathcal{H}_r$, a SVGD update of the samples $w_1^0, \ldots, w_N^0$ from $\pi_0(w)$ leads to

$$w_m^{\ell+1} = w_m^\ell + \epsilon_l \hat{\phi}_\ell^{r,*}(w_m^\ell), \quad m = 1, \ldots, N, \ell = 0, 1, \ldots, \tag{25}$$

with a step size $\epsilon_l$ and an approximate steepest direction by sample average approximation (SAA)

$$\hat{\phi}_\ell^{r,*}(w_m^\ell) = \frac{1}{N} \sum_{n=1}^{N} \nabla_{w_n^\ell} \log \pi(w_n^\ell) k^r(w_n^\ell, w_m^\ell) + \nabla_{w_n^\ell} k^r(w_n^\ell, w_m^\ell). \tag{26}$$

The kernel $k^r$ can be specified as in (9), i.e.,

$$k^r(w, w') = \exp\left(-\frac{||w - w'||_2^2}{h}\right). \tag{27}$$

To account for data impact in different directions $\psi_1, \ldots, \psi_r$ informed by the eigenvalues of (11), we propose to replace $||w - w'||_2^2$ in (27) by $(w - w')^T(\Lambda + I)(w - w')$ with $\Lambda = \mathrm{diag}(\lambda_1, \ldots, \lambda_r)$ for the likelihood and $I$ for the prior.

The following theorem, proved in Appendix A, gives $\nabla_w \log \pi(w)$ and the connection between pSVGD for the coefficient $w$ and SVGD for the projected parameter $P_r x$ under certain conditions.

**Theorem 1.** *The gradient of the posterior $\pi$ in (20) is given by*

$$\nabla_w \log \pi(w) = \Psi_r^T \left( \frac{\nabla_x g(P_r x)}{g(P_r x)} + \frac{\nabla_x p_0^r(P_r x)}{p_0^r(P_r x)} \right). \tag{28}$$

*Moreover, with the kernel $k^r(\cdot, \cdot) : \mathbb{R}^r \times \mathbb{R}^r \to \mathbb{R}$ defined in (27) and $k(\cdot, \cdot) : \mathbb{R}^d \times \mathbb{R}^d \to \mathbb{R}$ defined in (9), if $p_0^r(P_r x) = p_0(P_r x)$, for example $p_0$ is Gaussian, we have the equivalence of the projected transport map $T^r$ for the coefficient $w$ and the transport map $T$ for the projected parameter $P_r x$, as*

$$T^r(w) = \Psi_r^T T(P_r x). \tag{29}$$

*In particular, we have*

$$\nabla_w \log \pi(w) = \Psi_r^T \nabla_x \log p_r(P_r x). \tag{30}$$

**Remark 1.** *We can compute the gradient of the log-posterior $\log_w \log \pi(w)$ either by (28) or by the simplified formula (30) if $\pi_0(w) = p_0(P_r x)$, which is satisfied for $p_0$ such that its marginal for $P_r x$ is the same as the anchored density $p_0(P_r x + \xi)$ at $\xi = 0$. Meanwhile, Gaussian priors satisfy this condition since $p_0(P_r x + \xi) = p_0(P_r x)p_0(\xi)$ where both $p_0(P_r x)$ and $p_0(\xi)$ are Gaussian densities.*

### 3.3 Practical algorithms

Sampling from the projected density $p_r^*(x)$ defined in (13) for the optimal profile function $g^*$ in (16) involves, by the decomposition (21), sampling $w$ from the posterior $\pi$ by pSVGD and sampling $x^\perp$ from the conditional distribution with density $p_0^\perp(x^\perp|\Psi_r w)$. The sampling is impractical because of two challenges: (1) Both $p_0^\perp(x^\perp|\Psi_r w)$ and $g^*$ in (16) involve high-dimensional integrals. (2) The matrix $H$ defined in (10) for the construction of the basis $\psi_1, \ldots, \psi_r$ involves integration w.r.t. the posterior distribution of the parameter $x$. However, drawing samples from the posterior to evaluate the integral turns out to be the central task of the Bayesian inference.

The first challenge can be practically addressed by using the property that the posterior distribution only effectively differs from the prior in the dominant subspace $X_r$, or that the variation of likelihood $f$ in the complement subspace $X_\perp$ is negligible. Therefore, for any sample $x_n^0$ drawn from the prior $p_0$, $n = 1, \ldots, N$, we compute $x_n^\perp = x_n^0 - P_r x_n^0$ and freeze it for given $P_r$ as a sample from $p_0^\perp(x^\perp|P_r x)$. Moreover, at sample $x_n^\ell$ we approximate the optimal profile function $g^*$ in (16) as

$$g^*(P_r x_n^\ell) \approx f(P_r x_n^\ell + x_n^\perp), \tag{31}$$

which is equivalent to using one sample $x_n^\perp$ to approximate the integral (16) because the variation of $f$ in the complement subspace $X_\perp$ is small. This is used in computing $\nabla_w \log \pi(w)$ in (28).

Given the projector $P_r$ with basis $\Psi_r$, we summarize the pSVGD transport of samples in Algorithm 1. In particular, by leveraging the property that the samples can be updated in parallel, we implement

---

**Algorithm 1** pSVGD in parallel

---

1: **Input:** samples $\{x_n^0\}_{n=1}^N$ in each of $K$ cores, basis $\Psi_r$, maximum iteration $L_{\max}$, tolerance $w_{\mathrm{tol}}$.
2: **Output:** posterior samples $\{x_n^*\}_{n=1}^N$ in each core.
3: Set $\ell = 0$, project $w_n^0 = \Psi_r^T x_n^0$, $x_n^\perp = x_n^0 - \Psi_r w_n^0$, and perform MPI_Allgather for $\{w_n^0\}_{n=1}^N$.
4: **repeat**
5:     Compute gradients $\nabla_{w_n^\ell} \log \pi(w_n^\ell)$ by (28) for $n = 1, \ldots, N$, and perform MPI_Allgather.
6:     Compute the kernel values $k^r(w_n^\ell, w_m^\ell)$ and their gradients $\nabla_{w_n^\ell} k^r(w_n^\ell, w_m^\ell)$ for $n = 1, \ldots, NK$, $m = 1, \ldots, N$, and perform MPI_Allgather for them.
7:     Update samples $w_m^{\ell+1}$ from $w_m^\ell$ by (25) and (26) for $m = 1, \ldots, N$, with $NK$ samples used for SAA in (26), and perform MPI_Allgather for $\{w_m^0\}_{m=1}^N$.
8:     Set $\ell \leftarrow \ell + 1$.
9: **until** $\ell \geq L_{\max}$ or mean($||w_m^\ell - w_m^{\ell-1}||_2$) $\leq w_{\mathrm{tol}}$.
10: Reconstruct samples $x_n^* = \Psi_r w_n^\ell + x_n^\perp$.

---

a parallel version of pSVGD using MPI for information communication in $K$ processor cores, each with $N$ different samples, thus producing $M = NK$ different samples in total.

To construct the projector $P_r$ with basis $\Psi_r$, we approximate $H$ in (10) by

$$\hat{H} := \frac{1}{M} \sum_{m=1}^M \nabla_x \log f(x_m)(\nabla_x \log f(x_m))^T. \qquad (32)$$

where $x_1, \ldots, x_M$ are supposed to be samples from the posterior, which are however not available at the beginning. We propose to adaptively construct the basis $\Psi_r^\ell$ with samples $x_1^\ell, \ldots, x_M^\ell$ transported from the prior samples $x_1^0, \ldots, x_M^0$ by pSVGD. This procedure is summarized in Algorithm 2. We remark that by the adaptive construction, we push the samples to their posterior in each subspace $X_r^{\ell_x}$ spanned by (possibly) different basis $\Psi_r^{\ell_x}$ with different $r$ for different $\ell_x$, during which the frozen samples $x_n^\perp$ in Algorithm 1 are also updated at each step $\ell_x$ of Algorithm 2. We remark that different choices can be used for the step size function $\epsilon_l$. In the numerical experiments for structured models, we use a backtracking line search method with Armijo–Goldstein condition to look for the step size $\epsilon_l$, where the line search objective function is taken as the negative log-posterior function, see more details in the accompanying code at `https://github.com/cpempire/pSVGD`.

---

**Algorithm 2** Adaptive pSVGD in parallel

---

1: **Input:** samples $\{x_n^0\}_{n=1}^N$ in each of $K$ cores, $L_{\max}^x, L_{\max}^w, x_{\mathrm{tol}}, w_{\mathrm{tol}}$.
2: **Output:** posterior samples $\{x_n^*\}_{n=1}^N$ in each core.
3: Set $\ell_x = 0$.
4: **repeat**
5:     Compute $\nabla_x \log f(x_n^{\ell_x})$ in (32) for $n = 1, \ldots, N$ in each core, and perform MPI_Allgather.
6:     Solve (11) with $H$ approximated as in (32), with all $M = NK$ samples, to get bases $\Psi_r^{\ell_x}$.
7:     Apply the pSVGD Algorithm 1, i.e.,

$$\{x_n^*\}_{n=1}^N = \mathrm{pSVGD}(\{x_n^{\ell_x}\}_{n=1}^N, \Psi_r^{\ell_x}, L_{\max}^w, w_{\mathrm{tol}}).$$

8:     Set $\ell_x \leftarrow \ell_x + 1$ and $x_n^{\ell_x} = x_n^*$, $n = 1, \ldots, N$.
9: **until** $\ell_x \geq L_{\max}^x$ or mean($||x_m^{\ell_x} - x_m^{\ell_x-1}||_X$) $\leq x_{\mathrm{tol}}$.

---

## 4 Numerical Experiments

In this section, we present three Bayesian inference problems with high-dimensional parameters to demonstrate the accuracy of pSVGD compared to SVGD, and the convergence and scalability of pSVGD w.r.t. the number of parameters, samples, data points, and processor cores. A linear inference example, whose posterior is analytically given, is presented in Appendix B to demonstrate the accuracy of pSVGD. An application in COVID-19 to infer the time-dependent social distancing effect (which is high-dimensional after discretization in time) given hospitalized data is presented in

Appendix C. A more comprehensive application in COVID-19 to infer high-dimensional parameters and optimize mitigation strategies under uncertainty are given in [15, 9].

## 4.1 Conditional diffusion process

We consider a high-dimensional model that is often used to test inference algorithms in high dimensions, e.g., Stein variational Newton in [19], which is discretized from a conditional diffusion process

$$du_t = \frac{10u(1 - u^2)}{1 + u^2} dt + dx_t, \quad t \in (0, 1], \tag{33}$$

with zero initial condition $u_0 = 0$. The forcing term $(x_t)_{t \geq 0}$ is a Brownian motion, whose prior is Gaussian with zero mean and covariance $C(t, t') = \min(t, t')$. We use the Euler-Maruyama scheme with step size $\Delta t = 0.01$ for the discretization, which leads to dimension $d = 100$ for the discrete Brownian path $x$. We generate the data by first solving (33) at a true Brownian path $x_{\text{true}}$, and taking pointwise observation of the solution as $y = (y_1, \dots, y_{20})$ with $y_i = u_{t_i} + \xi_i$ for equispaced $t_1, \dots, t_{20}$ in $(0, 1]$ and additive noise $\xi_i \in N(0, \sigma^2)$ with $\sigma = 0.1$. We run SVGD and the adaptive pSVGD with line search for the learning rate, using $N = 128$ samples to infer the true parameter $x_{\text{true}}$, where the subspace for pSVGD is updated every 10 iterations. The results are displayed in Figure 1. From the left we can see a fast decay of eigenvalues of (11), especially when the iteration number $\ell$ becomes big with the samples converging to the posterior, which indicates the existence of an intrinsic low-dimensional subspace. From the right we can observe that pSVGD leads to samples at which the solutions are much closer to the noisy data as well as the true solution than that of SVGD. Moreover, the posterior mean of pSVGD is much closer to the true parameter with much tighter 90% confidence interval covering $x_{\text{true}}$ than that of SVGD.

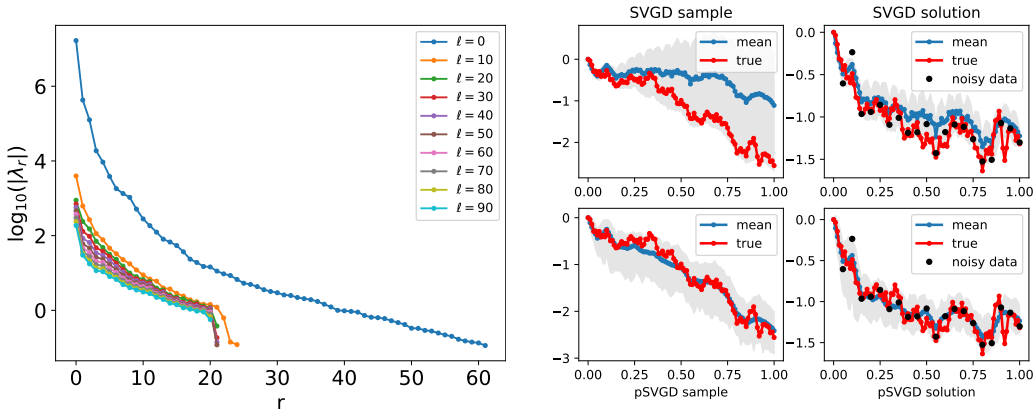

Figure 1: Left: Decay of the eigenvalues of (11) at different iteration numbers $\ell$. Right: SVGD (top) and pSVGD (bottom) samples $x$ and solutions $u$ at iteration $\ell = 100$, including the synthetic true, posterior mean, 90% confidence interval in shadow, and noisy data points.

## 4.2 Bayesian logistic regression

We consider Bayesian logistic regression for binary classification of cancer and normal patterns for mass-spectrometric data with $10,000$ attributes from https://archive.ics.uci.edu/ml/datasets/Arcene, which leads to $d = 10,000$ parameters (with i.i.d. uniform distribution as prior for Figure 2; Gaussian is also tested with similar results). We use 100 data samples for training and 100 for testing. We run pSVGD and SVGD with line search and 32 samples, with projection basis updated every 100 iterations. The results are shown in Figure 2. We can see a dramatic decay of the eigenvalues, which indicates an intrinsic dominant low dimensional subspace in which pSVGD effectively drives the samples to the posterior, and leads to more accurate prediction than SVGD. We remark that 32 samples in the estimate for the gradient information matrix in (32) are sufficient to capture the subspace since more samples lead to similar decay of eigenvalues as in Figure 2.

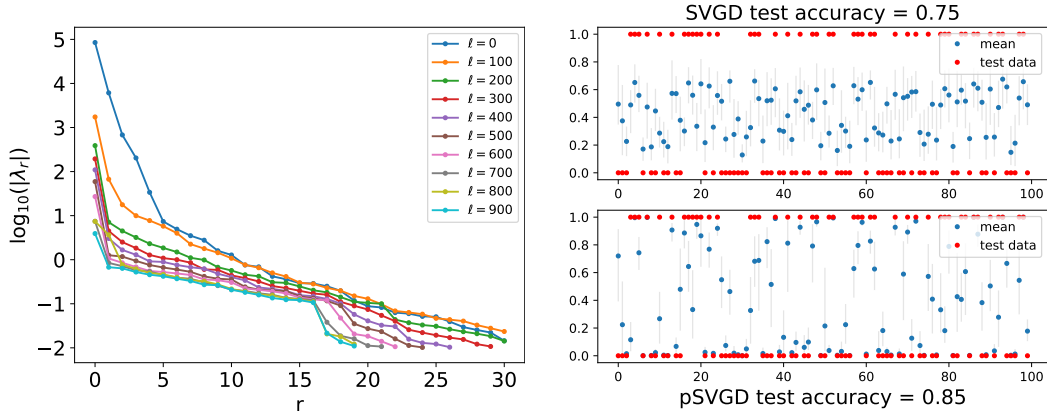

Figure 2: Left: Decay of the eigenvalues of (11) at different iteration numbers $\ell$. Right: SVGD (top) and pSVGD (bottom) test data and prediction at iteration $\ell = 1000$, including posterior mean, 90% confidence interval in shadow. The 85% test accuracy for pSVGD is the same as for SVM from the data source file. The training time for pSVGD is 201 seconds compared to 477 seconds for SVGD in a MacBook Pro Laptop (2019) with the processor of 2.4 GHz 8-Core Intel Core i9.

## 4.3 Partial differential equation

In this example we consider an elliptic partial differential equation model (widely used in various fields, e.g., inference for permeability in groundwater flow, thermal conductivity in material science, electrical impedance in medical imaging, etc.), with a simplified form as

$$-\nabla \cdot (e^{\mathrm{x}} \nabla \mathrm{u}) = 0, \quad \text{in } (0, 1)^2, \tag{34}$$

which is imposed with Dirichlet boundary conditions $\mathrm{u} = 1$ on the top boundary and $\mathrm{u} = 0$ on bottom boundary, and homogeneous Neumann boundary conditions on the left and right boundaries. $\nabla \cdot$ is a divergence operator, and $\nabla$ is a gradient operator. x and u are discretized by finite elements with piecewise linear elements in a uniform mesh of triangles of size $d$. $\mathrm{x} \in \mathbb{R}^d$ and $\mathrm{u} \in \mathbb{R}^d$ are the nodal values of x and u. We consider a Gaussian distribution for $\mathrm{x} \in \mathcal{N}(0, \mathcal{C})$ with covariance $\mathcal{C} = (-0.1\Delta + I)^{-2}$ (here $\Delta$ is Laplacian), which leads to a Gaussian distribution for $x \sim \mathcal{N}(0, \Sigma_x)$, where $\Sigma_x \in \mathbb{R}^{d \times d}$ is discretized from $\mathcal{C}$. We consider a parameter-to-observable map $h(x) = O \circ S(x)$, where $S : x \to u$ is a nonlinear discrete solution map of the equation (34), $O : \mathbb{R}^d \to \mathbb{R}^s$ is a pointwise observation map at $s = 7 \times 7 = 49$ points equally distributed in $(0, 1)^2$. We consider an additive 5% noise $\xi \sim \mathcal{N}(0, \Sigma_\xi)$ with $\Sigma_\xi = \sigma^2 I$ and $\sigma = \max(|Ou|)/20$ for data $y = h(x) + \xi$.

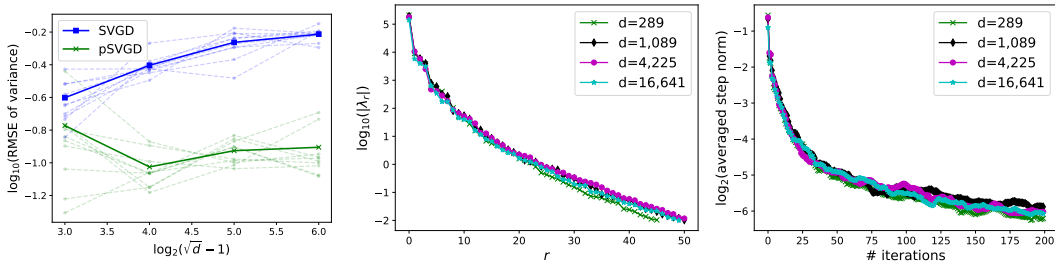

Figure 3: Left: RMSE of pointwise sample variance in $L_2$-norm, with dimension $d = (2^n + 1)^2$, $n = 3, 4, 5, 6$. Middle: Scalability w.r.t. $d$ by decay of eigenvalues $\lambda_r$ w.r.t. $r$. Right: decay of the averaged step norm $\text{mean}_m ||w_m^{\ell+1} - w_m^\ell||_2$ w.r.t. the number of iterations for different dimension $d$.

We use a DILI-MCMC algorithm [18] to generate $10,000$ effective posterior samples and use them to compute a reference sample variance. We run SVGD and the adaptive pSVGD (with $\lambda_{r+1} < 10^{-2}$) using 256 samples and 200 iterations for different dimensions, both using line search to seek the step size $\epsilon_\ell$. The comparison of accuracy can be observed in the left of Figure 3 by the root-mean-square-error (RMSE) of pointwise sample variance in $L_2$-norm. We can see that SVGD samples fail

to capture the posterior distribution in high dimensions and become worse with increasing dimension, while pSVGD samples represent the posterior distribution well, measured by sample variance, and the approximation remains accurate with increasing dimension. The accuracy of pSVGD can be further

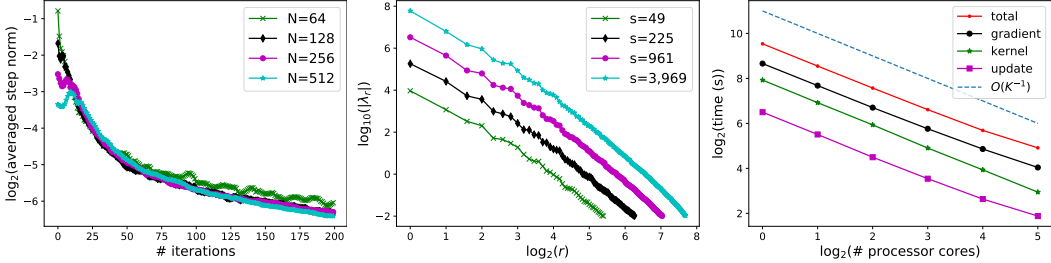

Figure 4: Scalability w.r.t. the number of (1) samples $N$ by decay of the averaged step norm (left), (2) data points $s$ by decay of eigenvalues (middle), and (3) processor cores $K$ by decay of CPU time (for gradient including eigendecompostion (11), kernel, sample update, total) of pSVGD (right).

demonstrated by the significant decay (about 7 orders of magnitude) of the eigenvalues for different dimensions in the middle of Figure 3. Only about 50 dimensions (with small relative projection error, about $\mathcal{E}_r < 10^{-6}$, committed in the posterior by (15)) are preserved out of from 289 to 16,641 dimensions, representing over $300\times$ dimension reduction for the last case. The similar decays of the eigenvalues $\lambda_r$ in the projection rank $r$ and the averaged step norm $\text{mean}_m ||w_m^{\ell+1} - w_m^\ell||_2$ in the number of iterations shown in the left of Figure 3 imply that pSVGD is scalable w.r.t. the parameter dimension. Moreover, the similar decays for different sample size $N = 64, 128, 256, 512$ in the left of Figure 4 demonstrate that pSVGD is scalable w.r.t. the number of samples $N$. Furthermore, as displayed in the middle of Figure 4, with increasing number of i.i.d. observation data points $s = 7^2, 15^2, 31^2, 63^2$ in a refined mesh of size $d = 17^2, 33^2, 65^2, 129^2$, the eigenvalues decay at almost the same rate with similar relative projection error $\mathcal{E}_r$, and lead to similar reduction $d/r$ for $r$ such that $\lambda_{r+1} < 10^{-2}$, which implies weak scalability of pSVGD w.r.t. the number of data points. Lastly, from the right of Figure 4 by the nearly $O(K^{-1})$ decay of CPU time we can see that pSVGD achieves strong parallel scalability (in computing gradient, kernel, and sample update) w.r.t. the number of processor cores $K$ for the same work with $KN = 1024$ samples.

## 5    Conclusions

We proposed a new algorithm — pSVGD for Bayesian inference in high dimensions to tackle the critical challenge of the curse of dimensionality. The projection error committed in the posterior can be bounded by the truncated (fast decaying) eigenvalues. We proved that pSVGD for the coefficient is equivalent to SVGD for the projected parameter under suitable assumptions. We demonstrated that pSVGD overcomes the curse of dimensionality for several high-dimensional Bayesian inference problems. In particular, we showed that pSVGD is scalable w.r.t. the number of parameters, samples, data points, and processor cores for a widely used benchmark problem in various scientific and engineering fields, which is crucial for solving high-dimensional and large-scale inference problems.

## Broader Impact

The proposed algorithm applies to high-dimensional Bayesian inference problems whose posterior effectively differs from the prior in a low-dimensional subspace discovered by the gradient information matrix, which is generally the case due to the fundamental property of the ill-posedness or over-parametrization of the inference problems. As one example, we applied the algorithm to Bayesian inference of the COVID-19 pandemics, which is expected to bring impact on learning the spread of the virus. However, for intrinsically high-dimensional problems, e.g., wave propagation in high frequency, a direct application of the proposed algorithm may not effectively alleviate the curse of dimenaionlity. It needs to be further extended by exploiting other properties such as sparsity, conditional independence, low-dimensionality with map transformation, etc. Impact on ethical aspects and future societal consequences may not be applicable here.

## Funding Disclosure

This work was partially funded by the Department of Energy, Office of Science, Office of Advanced Scientific Computing Research, Mathematical Multifaceted Integrated Capability Centers (MMICCS) program under award DE-SC0019303; the Simons Foundation under award 560651; and the National Science Foundation, Division of Mathematical Sciences under award DMS-2012453. Both authors have no competing interests with other entities.

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
