[Supplementary Material]

# A  Proof of Theorem 1

*Proof.* We first show (28). By definition of $\pi(w)$ in (20), we have

$$\nabla_w \log \pi(w) = \nabla_w \log(g(\Psi_r w)) + \nabla_w \log(\pi_0(w)). \tag{35}$$

For the second term, by definition of $\pi_0$ in (19), we have

$$\nabla_w \log(\pi_0(w)) = \nabla_w \log p_0^r(\Psi_r w) = \frac{\nabla_w p_0^r(\Psi_r w)}{p_0^r(\Psi_r w)}, \tag{36}$$

where by definition of $p_0^r(\Psi_r w)$ in (17) we have

$$\nabla_w p_0^r(\Psi_r w) = \int_{X_\perp} \nabla_w p_0(\Psi_r w + \xi) d\xi = \int_{X_\perp} \Psi_r^T \nabla_x p_0(\Psi_r w + \xi) d\xi = \Psi_r^T \nabla_x p_0^r(\Psi_r w). \tag{37}$$

The first term $\nabla_w \log(g(\Psi_r w))$ in (35) can be derived similarly by the definition of $g$ in (16) and (17).

Next, we proceed to prove the equivalence (29) and (30) at the first step $\ell = 0$. Then the equivalence for steps $\ell > 0$ follows by induction. By the parameter decomposition $x = x^r + x^\perp$ with $x^r = P_r x$, we denote $\eta_0$ and $\eta$ as the prior and posterior for the projected parameter $x^r$, given by

$$\eta_0(x^r) = p_0(P_r x) \text{ and } \eta(x^r) = p_r(P_r x), \tag{38}$$

where $p_r$ is the projected density defined in (13) with optimal profile function $g = g$ given in (16). Equivalently, by the property of the projection $P_r P_r x = P_r x$, we have

$$\eta(x^r) = \frac{1}{Z_r} g(x^r) \eta_0(x^r). \tag{39}$$

We can write the transport map (4) for the projected parameter $x^r$ in the steepest direction $\varphi_0$ as

$$T(x^r) = x^r + \epsilon \varphi_0(x^r), \tag{40}$$

where $\varphi_0$ is given by

$$\varphi_0(\cdot) = \mathbb{E}_{x^r \sim \eta_0}[\mathcal{A}_\eta \kappa(x^r, \cdot)], \tag{41}$$

with the kernel $\kappa(x^r, \tilde{x}^r) = k(P_r x, P_r \tilde{x})$ for any $x, \tilde{x} \in \mathbb{R}^d$ and the Stein operator

$$\mathcal{A}_\eta \kappa(x^r, \cdot) = \nabla_{x^r} \log \eta(x^r) \kappa(x^r, \cdot) + \nabla_{x^r} \kappa(x^r, \cdot). \tag{42}$$

By definition of the kernel in (9), we have

$$
\begin{aligned}
k(P_r x, P_r \tilde{x}) \\
= \exp\left(-\frac{1}{h}(P_r x - P_r \tilde{x})^T (P_r x - P_r \tilde{x})\right) \\
= \exp\left(-\frac{1}{h}(w - \tilde{w})^T \Psi_r^T \Psi_r (w - \tilde{w})\right) \\
= \exp\left(-\frac{1}{h}\|w - \tilde{w}\|_2^2\right)
\end{aligned}
\tag{43}
$$

where we used the relation $P_r x = \Psi_r w$ and $P_r \tilde{x} = \Psi_r \tilde{w}$ in the second equality and the orthonormality $\Psi_r^T \Psi_r = I$ in the generalized eigenvalue problem (11) in the third. Therefore, by definition (27), we have

$$k^r(w, \tilde{w}) = \kappa(x^r, \tilde{x}^r). \tag{44}$$

Moreover, for the gradient of the kernel we have

$$
\begin{aligned}
\nabla_{x^r} \kappa(x^r, \tilde{x}^r) &= -\frac{2}{h} \kappa(x^r, \tilde{x}^r)(x^r - \tilde{x}^r) \\
&= -\frac{2}{h} \kappa(x^r, \tilde{x}^r) \Psi_r (w - \tilde{w}).
\end{aligned}
\tag{45}
$$

On the other hand, we have

$$\nabla_w k^r(w, \tilde{w}) = -\frac{2}{h} k^r(w, \tilde{w})(w - \tilde{w}), \tag{46}$$

which yields

$$\nabla_w k^r(w, \tilde{w}) = \Psi_r^T \nabla_{x^r} \kappa(x^r, \tilde{x}^r). \tag{47}$$

For the posterior $\eta$ defined in (38), we have

$$\nabla_{x^r} \log \eta(x^r) = \frac{\nabla_{x^r}(g(x^r)\eta_0(x^r))}{g(x^r)\eta_0(x^r)}, \tag{48}$$

while for the posterior $\pi$ defined in (20), we have

$$\nabla_w \log \pi(w) = \frac{\nabla_w(g(\Psi_r w)\pi_0(w))}{g(\Psi_r w)\pi_0(w)}. \tag{49}$$

By chain rule, it is straightforward to see that

$$\nabla_w g(\Psi_r w) = \Psi_r^T \nabla_{x^r} g(x^r). \tag{50}$$

Under assumption $\pi_0(w) = p_0(P_r x)$ in Theorem 1, and $p_0(P_r x) = \eta_0(x^r)$ by definition (38), we have

$$\nabla_w \pi_0(w) = \Psi_r^T \nabla_{x^r} \eta_0(x^r). \tag{51}$$

Therefore, combining (50) and (51), we have

$$\nabla_w \log \pi(w) = \Psi_r^T \nabla_{x^r} \log \eta(x^r). \tag{52}$$

To this end, we obtain the equivalence of the Stein operators

$$\mathcal{A}_\pi k^r(w, \tilde{w}) = \Psi_r^T \mathcal{A}_\eta \kappa(x^r, \tilde{x}^r) \tag{53}$$

for $x^r = \Psi_r w$ and $\tilde{x}^r = \Psi_r \tilde{w}$. Since the prior densities $\eta_0(x^r) = \pi_0(w)$, we have the equivalence

$$\mathbb{E}_{w \sim \pi_0}[\mathcal{A}_\pi k^r(w, \tilde{w})] = \Psi_r^T \mathbb{E}_{x^r \sim \eta_0}[\mathcal{A}_\eta \kappa(x^r, \tilde{x}^r)], \tag{54}$$

which concludes the equivalence of the transport map (29) by $w = \Psi_r^T x^r$ with the same $\epsilon$ at $\ell = 0$. Moreover, by induction we have

$$T_\ell^r(w^\ell) = \Psi_r^T T_\ell(P_r x^\ell), \tag{55}$$

which concludes.

$\square$

## B  A linear inference problem

This example is presented to test the accuracy of the proposed algorithm with analytically given posterior distribution for a linear inference problem. We consider a linear parameter-to-observable map $A : \mathbb{R}^d \to \mathbb{R}^s$, which is given by

$$Ax = O \circ Bx, \tag{56}$$

where $B : x \to u$ is a linear discrete solution map of the diffusion reaction equation ($\Delta$ is the Laplace operator)

$$-\Delta u + u = x, \quad \text{in } (0, 1), \tag{57}$$

with boundary condition $u(0) = 0$ and $u(1) = 1$, which is solved by a finite element method. The continuous parameter x and solution u are discretized by finite elements with piecewise linear elements in a uniform mesh of size $d$. $x \in \mathbb{R}^d$ and $u \in \mathbb{R}^d$ are the nodal values of x and u. The parameter x is assumed to follow a Gaussian distribution $\mathcal{N}(0, \mathcal{C})$ with covariance $\mathcal{C} = (-0.1\Delta + I)^{-1}$, which leads to a Gaussian parameter $x \sim \mathcal{N}(0, \Sigma_x)$, with covariance $\Sigma_x \in \mathbb{R}^{d \times d}$ as a discretization of $\mathcal{C}$.

$O : \mathbb{R}^d \to \mathbb{R}^s$ in (56) is an observation map that take $s$ components of $u$ that are equally distributed in $(0, 1)$. For $s = 15$, we have $Ou = (u(1/16), \dots, u(15/16))^T$. We assume an additive 1% Gaussian noise $\xi \sim \mathcal{N}(0, \Sigma_\xi)$ with $\Sigma_\xi = \sigma^2 I$ and $\sigma = \max(|Ou|)/100$ for data

$$y = Ax + \xi, \tag{58}$$

then the likelihood function is given by

$$f(x) = \exp\left(-\frac{1}{2}||y - Ax||^2_{\Sigma_\xi^{-1}}\right). \tag{59}$$

Because of the linearity of the inference problem, the posterior of $x$ is also Gaussian $\mathcal{N}(x_{\text{MAP}}, \Sigma_y)$ with the MAP point $x_{\text{MAP}} = \Sigma_y A^T \Sigma_\xi^{-1} y$ and covariance

$$\Sigma_y = (A^T \Sigma_\xi^{-1} A + \Sigma_x^{-1})^{-1}. \tag{60}$$

We run SVGD and pSVGD (projection with $r = 8$ basis functions and $\lambda_9 < 10^{-4}$) with 256 samples and 200 iterations for different dimensions, both using line search to seek the step size $\epsilon_\ell$. The RMSE (of 10 trials and their average) of the samples variances compared to the ground truth (60) are shown in Figure 5, which indicates that SVGD deteriorates with increasing dimension while pSVGD performs well for all dimensions.

Figure 5: RMSE of pointwise sample variance in $L_2$-norm, with 256 samples, SVGD and pSVGD both terminated at $\ell = 200$ iterations, parameter dimension $d = 2^n + 1$, with $n = 4, 6, 8, 10$.

## C  Application to COVID-19 modeling

Social distancing has played a key role in flattening the curve of the spread of COVID-19. In this example, we apply pSVGD to infer a time-dependent parameter that represents the reduction in contacts due to social distancing, given observation data. We consider a compartmental model with 8 compartments for the modeling of the transmission and outcome of COVID-19, as illustrated by the diagram in Figure 6. The corresponding model is given by the system of ordinary differential equations

Figure 6: Diagram of a compartmental epidemic model with 8 compartments for modeling of transmission and outcome of infectious diseases such as COVID-19.

$$C_E(t) = (1 - \alpha(t))(1 - q)\frac{I(t)}{N} + (1 - \alpha(t))\frac{A(t)}{N},$$

$$C_Q(t) = (1 - \alpha(t))q\frac{I(t)}{N},$$

$$\frac{dS(t)}{dt} = -\beta C_E(t)S(t) - \beta C_Q(t)S(t),$$

$$\frac{dE(t)}{dt} = \beta C_E(t)S(t) - \tau\sigma E(t) - (1 - \tau)\sigma E(t),$$

$$\frac{dQ(t)}{dt} = \beta C_Q(t)S(t) - \rho\eta_Q Q(t) - (1 - \rho)\gamma_Q Q(t),$$

$$\frac{dA(t)}{dt} = (1 - \tau)\sigma E(t) - \gamma_A A(t),$$

$$\frac{dI(t)}{dt} = \tau\sigma E(t) - \pi\eta_I I(t) - (1 - \pi)\gamma_I I(t),$$

$$\frac{dH(t)}{dt} = \pi\eta_I I(t) + \rho\eta_Q Q(t) - \nu\mu H(t) - (1 - \nu)\gamma_H H(t),$$

$$\frac{dR(t)}{dt} = \gamma_A A(t) + (1 - \pi)\gamma_I I(t) + (1 - \nu)\gamma_H H(t) + (1 - \rho)\gamma_Q Q(t),$$

$$\frac{dD(t)}{dt} = \nu\mu H(t).$$

(61)

In this model, $\alpha(t) \in [0, 1]$ represents the effective contact reduction due to social distancing at time $t$, i.e., the fractional reduction in contacts relative to no social distancing. This is the time-varying parameter we infer. The other parameters are defined as folows: $N$ is the total population for a given region; $\beta$ is transmission rate; $q$ is quarantined rate; $\sigma$ is latency rate; $\eta_I, \eta_Q$ are hospitalized rates; $\gamma_A, \gamma_I, \gamma_Q, \gamma_H$ are recovery rates from each compartment; $\mu$ is deceased rate; and $\tau, \rho, \pi, \nu$ are the proportions of cases going from $E$ to $I$, $Q$ to $H$, $I$ to $H$, and $H$ to $D$. We assume these parameters are scalar and do not change over time. For the observational data, we use the daily number of currently hospitalized patients $H$ (7 days' moving average) in New York state over the period March 16–June 4, 2020 obtained from `https://github.com/COVID19Tracking`, and assume that the observational noise is i.i.d. $N(0, 1)$ for the logarithm of the daily number to create the likelihood function.

First, we solve a deterministic optimization problem to infer all parameters by minimizing the data misfit between the logarithm of the predicted and observed daily numbers of hospitalized patients. Then we freeze all the parameters at their inferred values, except for the social distancing function $\alpha(t)$. We assume that

$$\alpha(t) = \frac{1}{2}(\tanh(x(t)) + 1)$$

is a stochastic process with Gaussian process $x(t) \sim \mathcal{N}(\hat{x}(t), \mathcal{C})$, where $\hat{x}(t) = \text{arctanh}(2\hat{\alpha}(t) - 1)$ is evaluated at the deterministic optimal values of the social distancing function $\hat{\alpha}(t)$ obtained from the deterministic optimization problem, and $\mathcal{C} = -(\delta \triangle_t)^{-1}$ with Laplacian operator $\triangle_t$ and a scaling parameter $\delta = 10$. After discretization in time with step of one day over 96 days, we obtain a discrete parameter $x = (x_1, \ldots, x_d)$ of dimension $d = 96$. We run pSVGD and SVGD with line search, 128 samples with 8 samples on each of 16 processor cores, and update the bases for pSVGD every 10 iterations for a total of 200 iterations. The results are shown in Figure 7. We can observe a fast decay of eigenvalues and a low intrinsic dimension. The bottom two figures display the samples, their means, and 90% confidence interval of the reduction factor $\alpha$ for social distancing and the number of hospitalized cases by SVGD (top) and pSVGD (bottom). We can observe from the right of Figure 7 that pSVGD provides much more accurate prediction of the data (the number of hospitalized patients) with tighter confidence interval than SVGD. Meanwhile, the 90% confidence interval of pSVGD covers the deterministic optimal value, which is also close to the posterior mean of pSVGD, while that of SVGD does not. The mean of the reduction factor $\alpha$ for SVGD is nearly 1 for a period of time, which corresponds to complete shutdown without any transmission, which is not realistic. We remark that the sharp increase of the social distancing factor in late March/early April is due to the implemented mitigation measure of lockdown, which demonstrate the efficacy of pSVGD for compartmental model based Bayesian inference, which also quantifies the uncertainty of the parameters. Inference of other parameters with quantified uncertainty, e.g., time-dependent infection hospitalization ratio, hospitalization fatality ratio, given data of confirmed, hospitalized, and deceased cases, can be found in [15], which is used for stochastic optimal mitigation design under parameter uncertainty in [9].

## D   Remarks on computational cost of pSVGD and pSVN

In comparison with pSVN [14], pSVGD uses only gradient information of the log-likelihood, which is available for many models, while pSVN requires Hessian information, which is challenging for complex models and codes in practical applications. Moreover, computing the Hessian in pSVN could be much more demanding — (1) evaluating the full Hessian would be $d$ times more expensive than evaluating the gradient for $d$-dimensional parameter inference problem, (2) evaluating a Gaussian–Newton approximation of the Hessian requires Jacobian, which is $s$ times more expensive for $s$-dimensional data, (3) a low-rank approximation with rank $r$ of the Hessian is $O(r)$ times more expensive by efficient randomized algorithm. In contrast, the gradients of the log-likelihood at each sample used by pSVGD are already computed in SVGD, which are the dominant computational cost.

Figure 7: Top: Decay of the eigenvalues of (11) at different iteration numbers $\ell$. Bottom-left: samples of $\alpha$ (contact reduction due to social distancing): posterior mean in blue with 90% confidence interval in grey and deterministic optimal $\hat{\alpha}$ in black. Bottom-right: the number of hospitalized cases: posterior mean in blue with 90% confidence interval in grey compared to reported data for New York state in black. The results are at iteration $\ell = 200$.