[Reviews · NeurIPS 2020]

Review 1

Summary and Contributions: In many Bayesian posterior sampling problems, the posterior and the prior differs only on a low dimension linear subspace. This linear subspace contains most of the information regarding the posterior distribution, given the prior distribution. To tackle the curse of dimensionality, this paper proposes Projected Stein Variational Gradient Descent (pSVGD) which can be seen as SVGD implemented on a well chosen subspace of the parameter space. More precisely, the projection formulas to implement pSVGD are proven, approximate implementations of pSVGD are given and extensive numerical experiment on several Bayesian inference problems are given.

Strengths: The main theoretical results of the paper are exact formulas for the quantities needed to implement SVGD on a subspace. The authors also provide intuitions on the chosen subspace as well as and several numerical examples in which this subspace contains most of the relevant information. The "information" is represented as some generalized eigenvalues of the gradient information matrix and the dimension of the relevant linear subspace is the number of "large" generalized eigenvalues. The generalized eigenvectors are involved in the formulas proven. The paper is mainly empirical. pSVGD cannot be implemented in closed form because the quantities needed to implement pSVGD depend on some unknowns, such as integrals w.r.t. the posterior distribution. Therefore, approximate implementations of pSVGD are proposed, which are derived by making successive approximations. These algorithms can be implemented in parallel. In the first approximate implementation, the generalized eigenvectors are assumed to be known, and in the second one, the generalized eigenvectors are updated online. The numerical experiments do show that several Bayesian inference problems exhibit a low dimensional subspace containing most of the information about the posterior distribution. This is shown via the fast decay of the generalized eigenvalues. Moreover, pSVGD show some superiority over SVGD on some prediction/regression tasks in practice.

Weaknesses: Regarding the numerical experiements, I don't understand how pSVGD should work better than SVGD. Is it in terms of the number of iterations needed to achieve epsilon accuracy of some criterion? Or is it that working on a subspace is less computationally demanding? This is not clear to me what are the advantages of pSVGD over SVGD. The experiments show some improvements, but this is not supported by any theory. I would expected that we lose something by implementing pSVGD instead of SVGD. What do we lose? Moreover, I didn't know that SVGD suffers the curse of dimensionality (see l.84). Can the author explain this point? Finally, the paper's goal is to tackle the curse of dimensionality for SVGD but no result is proven in terms of complexity w.r.t. to the dimension. I know that the complexity of SVGD is an open problem, but can the authors give some hints to understand how the SVGD suffers the curse of dimensionality and how pSVGD improves that? Generalized eigenvalues. Do these generalized eigenvalues always exist? I understand the intuition l.100, but can the authors formalize these points? I feel that the paper is not enough theoretically grounded. Figure 3. Why is the variance of the samples important here? If the target distribution has a high variance then the samples should also have a high variance, right?

Correctness: In my opinion the paper is not enough theoretically grounded and does not answer the questions related to the dimension. However, the questions related to the implementation of pSVGD (i.e., the proofs) look correct. The experiment are relevant to the messages of the paper.

Clarity: The theoretical part of the paper is well written and easy to follow. Reading the practical part (starting from Section 3.3) is more difficult because of several undefined concepts. For instance, Algorithms 1 and 2 rely on some MPI subroutines that I don't know. I wanted to see the formula for the fundamental quantity nabla log pi (w) written somewhere *in the algorithm* but it is not. This makes the algorithm difficult to implement for the reader (I had to compute the formula by myself). Moreover, some abbreviations are not defined, for instance SAA, RMSE. Same in section 4 (numerical experiements) where many concepts are implicitly defined. Example: Sample vs solution in Figure 1, \Delta (l. 211), etc. The definition of s is changed from l. 213 to Figure 4. The linesearch of pSVGD which is used in every numerical experiments is not explained. Minor comment: - l. 228. "right" should be "left"? - The training time is given l. 201 but what machine is used ? - l. 73. Why is T invertible?

Relation to Prior Work: Related works are acknowledged. The closest work is projected Stein Variational Newton. pSVGD can be seen as an analogue to pSVN using gradient descent instead of Newton's algorithm. Some other differences that goes beyond this analogy are given. For instance, an equivalence proven in Theorem 1 has no analogue for pSVN.

Reproducibility: Yes

Additional Feedback: -------------------------------------------------------------------------------------- I thank the authors for their answers. My concern about their work not being theoretically grounded was not answered, but I understand that the authors made an interesting empirical contribution by proposing pSVGD. I also understand that the literature on SVGD is rather empirical. Therefore, I am raising my score.


Review 2

Summary and Contributions: *EDIT* Given the author responses to my questions and their willingness to update the clarity of the paper, I am strongly in favour of its acceptance now and changed my score accordingly. This paper proposes a method, pSVGD, to compute parameters inference in high-dimensional nonlinear Bayesian problems, overcoming many challenges due to the curse of dimensionality. The method exploits the fact that the posterior of the parameters only differs from the prior in a low-dimensional subspace of the parameters space. Therefore, pSVGD projects the parameters to a low-dimensional subspace. This subspace is found using the gradient of the log-likelihood. Moreover, the authors prove that their method is equivalent to using the transport of the projected parameters under certain conditions, and that the bound for the projection error is lower than of pSVN (concurrent method). They also provide practical and scalable parallel algorithms to compute the different integrals needed in their method. Experiments are performed on different challenging nonlinear Bayesian inference problem to show the relevance of pSGVD over SGVD. Code is available for their method.

Strengths: Strengths: * This paper is very sound (claims are proven and empirical evaluation is performed with care). * The proposed method, albeit relying on a simple observation, can have high significance to the Bayesian community. * The proposed method is a novel modification of the SVGD algorithm. * I believe this paper to be of good significance to the NeurIPS community.

Weaknesses: *EDIT* Given the author responses to my questions and their willingness to update the clarity of the paper, I am strongly in favour of its acceptance now and changed my score accordingly. Weaknesses: The main weakness of this paper is its lack of clarity (see below).

Correctness: Yes (but I did not check the proof in Appendix A).

Clarity: The main weakness of this paper is the lack of clarity, the paper is very hard to read. Little intuition and perspective is given on the method (for example, there is no analysis of the consequence of Theorem 1) and many notations are employed. This is reason why I gave the current score, and I am more than willing to upgrade my score to full accept if the authors clarify Section 3. I appreciated that the author made the effort to present SVGD before presenting their method. I would recommend the authors to greatly improve the clarity of their paper by: * Clarifying the different steps in Section 3 as an introduction to the Section. For example, in the current submission the section starts with the introduction of H without any intuition on its purpose (if I understood correctly, it it used to find the low-dimensional subspace on which projection is made, by finding the eigenvalues \lambda_i). * Decrease the number of notations (\omega, \pi, g, etc.) introduced when possible.

Relation to Prior Work: Yes the authors clearly discuss their difference with SGVD and pSVN.

Reproducibility: Yes

Additional Feedback: Due to the lack of clarity of the paper, I have a few questions to the authors: * What is the computational cost of their method compared to SVGD in the experiments (a few values / plots would be relevant). * Would it be possible/easy to report pSVN performance for comparison ? * What is the value of r (or threshold on \lambda_r+1) used in the experiments when Test error are reported ? * Why don't you discuss the results in Appendix on the Covid data for example ? They could be a good proof for the broader impact of your work.


Review 3

Summary and Contributions: The paper proposes a novel method for high-dimensional nonlinear Bayesian inference problems. Specifically, the paper proposes projected Stein variational gradient descent, an extension of prior work that computes parameter posterior updates in a low-dimensional subspace to avoid the curse of dimensionality. The main contributions of the paper are the design, theoretical, and empirical analysis of the algorithm. In particular, the paper relates the proposed method to stein variational gradient descent for the projected parameters and provides a set of inference problems, where the proposed method outperforms alternative methods.

Strengths: ++++++++++++++++++++++++++++++++++++++++++++++++++++++++++++++ ++ Preface ++ I understand that reviews that claim that a method is not sufficiently novel or significant are often subjective and are difficult for authors to rebut. To make my review easier to engage with, I’m offering the following criteria along which I assess “significance” of a paper: (*i*) Does the paper offer a novel, non-obvious theoretical insight in the form of a proof or derivation? (*ii*) Does the paper propose a new method that leverages novel, non-trivial theoretical, conceptual, and/or empirical insights ? (*iii*) Does the paper present believable/verifiable results that solve a previously unsolved problem or that significantly improve upon existing methods. I will touch on these three criteria in my comments below and mark my comment accordingly. ++++++++++++++++++++++++++++++++++++++++++++++++++++++++++++++ ++ Relevance ++ Bayesian inference applied to a variety of problems is an active area of research and the paper under review proposes a novel algorithm for fast convergence to a posterior distribution in Bayesian inference problems. While the proposed method is still limited in the parameter dimension, it improves on related methods and makes stein variational gradient descent more practically relevant. As such, I believe the paper is of relevance to the NeurIPS community. ++++++++++++++++++++++++++++++++++++++++++++++++++++++++++++++ ++ Significance and novelty of contribution ++ (*i*) Does the paper offer a novel, non-obvious theoretical insight in the form of a proof or derivation? The paper proves that projected transport in the coefficient space and the transport (via Stein variational gradient descent) in the projected parameter space are, under certain conditions, equivalent. Furthermore, the paper shows that the projection error in the posterior can be bounded by the truncated eigenvalues of the empirical Fisher information. (*ii*) Does the paper propose a new method that leverages novel, non-trivial theoretical, conceptual, and/or empirical insights? One of the key contributions of the paper is the insight that the posterior only effectively differs from the prior in a relatively low-dimensional subspace when the inference problem is ill-posed or overparameterized. It uses this insight as a justification for projecting the parameters to a low-dimensional subspace, which makes it possible to circumvent the curse of dimensionality in Stein variational gradient descent. The proposed algorithm identifies this low-dimensional subspace via the largest eigenvalues of the empirical Fisher information matrix. The paper nicely leverages prior work on Stein variational gradient descent and uses simple, intuitively well-justified insights for reducing the curse of dimensionality in Bayesian inference problems, and I believe that the methodological contribution, while a direct and simple extension of prior work, is sufficiently significant to warrant publication. (*iii*) Does the paper present believable/verifiable results that solve a previously unsolved problem or that significantly improve upon existing methods. The empirical evaluation demonstrates that the proposed method does in fact scale Stein variational gradient descent to high-dimensional inference problems. See below under empirical claims. ++++++++++++++++++++++++++++++++++++++++++++++++++++++++++++++ ++ Soundness of theoretical claims ++ I checked the proof in the appendix and did not find any errors. ++++++++++++++++++++++++++++++++++++++++++++++++++++++++++++++ ++ Quality and soundness of empirical claims ++ The empirical investigation is insightful and covers a range of relevant inference problems. The different problem setups are explained clearly, and the presentation of the experimental results is clear and easy to understand. Importantly, the paper validates that the eigenvalues in fact decay rapidly, indicating that the proposed method transports the samples from the prior to the posterior effectively. ++++++++++++++++++++++++++++++++++++++++++++++++++++++++++++++

Weaknesses: ++++++++++++++++++++++++++++++++++++++++++++++++++++++++++++++ ++ Relevance ++ See under “Strengths” above. ++++++++++++++++++++++++++++++++++++++++++++++++++++++++++++++ ++ Significance and novelty of contribution ++ See under “Strengths” above. ++++++++++++++++++++++++++++++++++++++++++++++++++++++++++++++ ++ Soundness of theoretical claims ++ The paper claims that the proposed method does not require computation of the Hessian. However, it does require the computation of the empirical Fisher information, which is technically an approximation to the Hessian and also contains second-order information. How much more computationally efficient is it to compute the empirical Fisher information than to compute the Hessian? What is the actual computational saving of the proposed method compared to pSVN? ++++++++++++++++++++++++++++++++++++++++++++++++++++++++++++++ ++ Quality and soundness of empirical claims ++ While I believe that the empirical results included in the paper are insightful, it would have been nice to demonstrate the performance of the proposed method on a simple reinforcement learning or Bayesian deep learning problem. ++++++++++++++++++++++++++++++++++++++++++++++++++++++++++++++

Correctness: I checked the proof in the appendix and did not find any errors.

Clarity: ++++++++++++++++++++++++++++++++++++++++++++++++++++++++++++++ ++ Strengths ++ The paper is well-structured and well-written. The explanations are generally clear and easy to follow. The figures looked professional and were effective in communicating key insights and results. ++++++++++++++++++++++++++++++++++++++++++++++++++++++++++++++ ++ Weaknesses ++ Unless I’m mistaken, the approximation to H is just the empirical Fisher information. If so, it may be worth explicitly stating so. Minor issues: - Line 150: replace dash (--) by “that” - Line 151: “or the variation” -> “or that the variation” - Line 179: “We use Euler-Maruyama scheme” -> “We use the Euler-Maruyama scheme” - Line 195: “We use 100 data for training” -> “We use 100 data samples for training” - Line 199: “more accurate prediction than that of SVGD” -> “more accurate prediction than SVGD” ++++++++++++++++++++++++++++++++++++++++++++++++++++++++++++++

Relation to Prior Work: The paper provides a detailed list of references and engages with prior work on a high-level without getting into specific details of individual papers and instead refers to common characteristics among sets of papers (see, for example, the introduction and the first paragraph of Section 3). A more thorough discussion of the proposed method vis-a-vis other work would be welcome.

Reproducibility: Yes

Additional Feedback: I would appreciate it if the authors could answer my question regarding the the computational saving of the proposed method relative to, for example, pSVN (see my comment above) and why they did not include experiments on RL or Bayesian deep learning to make their proposed method more practically relevant for larger sections of the NeurIPS community. All in all, I believe that the contribution of the paper is significant, but I am not intimately familiar with the recent literature on Stein variational gradient descent, meaning that it is possible that my assessment is too positive. In any case, I will gladly change my score if the author response and/or the discussion with reviewers convinces me that my assessment was wrong. *************************** POST-REBUTTAL UPDATE *************************** I thank the authors for addressing my questions. I will keep my score as "7: A good submission; accept," since I don't believe the contribution of the paper warrants a higher score.


Review 4

Summary and Contributions: In this paper, the authors propose a new approximate algorithm for Bayesian inference for high dimensions. For the Bayesian inference setup, under the assumption that prior distribution differs from the posterior distribution only in a low dimensional subspace, the authors propose projected Stein Variational gradient descent algorithm. The main idea of this algorithm is similar to the Stein Variational Gradient Descent (SVGD) where the main difference is that instead of computing the transport map for the full d dimensional state space, authors propose to only compute the transport map for the low dimensional subspace with the basis given by the top r eigenvalues of the gradient information matrix while keeping the transportation map for the other d-r dimensional subspace as the identity map. Authors mention the previous theoretical work by Zahm et al. 2018 that give the justification for only using the subspace spanned by the top r eigenvectors of the gradient information matrix. The new approximate posterior is given by scaling the prior by the profile function which is the marginal likelihood of the subspace. The ideal implementation of the pSVGD is not feasible because a) computing profile function requires computing an integral and b) more importantly the gradient information matrix is not efficiently computable since it requires integration wrt the posterior. For the practical implementation of a) authors use single sample Monte Carlo estimate. For b), the authors propose a heuristic that adaptively computes the local approximation to the posterior distribution and uses this approximation of the posterior to compute the approximate gradient information matrix. Authors also give some simulation studies validating their approach.

Strengths: The problem discussed in the paper is very relevant for Bayesian inference in high dimensions and the authors propose a novel approach to perform approximate Bayesian inference in high dimensions. The ideas presented in this paper (to the best of my knowledge) are novel and give a new approximate inference algorithm for high dimensional Bayesian inference.

Weaknesses: The heuristic proposed in the paper that computes the gradient information matrix has no particular convergence guarantee. I would encourage authors to give experiments for a simple model where the one can compute by hand the matrix and validate how the approximation scheme is working in practice. Also, I would like to see a computational complexity analysis in terms of the dimensionality of the data for pSVGD and SVGD to compare the run time. This concern is also because the numerical experiments presented in the paper were on relatively small datasets. I would like to see how this scales in both number of data points and dimensionality of the data.

Correctness: To the best of my knowledge, the claims in the paper are correct. To evaluate the performance of their methods, authors run experiments that validate the findings.

Clarity: Overall the technical presentation of the paper is clear and easy to follow.

Relation to Prior Work: Tthe approximation used in the paper heavily relies on the theoretical work done by Zahm et al. 2018. I would like the authors to make that connection clear in the introduction as the only reference I found of their work was in eq (15).

Reproducibility: Yes

Additional Feedback:

[Author Response · NeurIPS 2020]

We are grateful to all the reviewers for their careful and constructive assessment of our work, and in particular thank for their helpful suggestions for its improvement. We appreciate the reviewers' recognition of our key contribution in addressing the longstanding challenge of the curse of dimensionality for high-dimensional Bayesian inference problems, as well as its high relevance and significant impact in NeurIPS community.

Below are our responses to the constructive criticisms, insightful questions, and helpful suggestions of each reviewer.

**To Reviewer 1:** We thank for your understanding of the key property of the low-dimensionality of the subspace in which the posterior differs from the prior, and our motivation in exploiting this property by adaptively projecting the parameters into carefully constructed subspaces to use SVGD. **We also appreciate your frank expression of not understanding the fundamental limitation of SVGD and the advantages of our pSVGD over SVGD for high-dimensional inference, which led to your different reject score in contrast to all the other accept scores**. SVGD is limited because the kernel given in (9) leads to severe degeneracy of the repulsive force of SVGD in high dimensions, which makes samples collapsing to the modes of the posterior, as observed in [31, 34]. This is demonstrated by the significantly inaccurate posterior sample variances with increasing dimensions in Figure 3 for nonlinear problems (compared to DILI MCMC reference) and in Figure 5 for linear problems (compared to the ground truth) in Appendix B. In contrast, the posterior sample variance of pSVGD samples is preserved to be much more accurate than SVGD with increasing nominal dimensions as shown in both Figure 3 and 5, because pSVGD extracts the essential information, i.e., by projecting parameters to the intrinsically low-dimensional subspaces regardless of the nominal dimensions. This is also demonstrated by the much smaller testing errors of pSVGD compared to SVGD in Figure 1, 2, and 7. The advantage is also (but less) about the number of iterations or computational complexity, even though pSVGD converges faster with less cost (because of optimization in low dimensions) than SVGD as demonstrated in Figure 2. Admittedly, pSVGD is more involved to implement than SVGD. By definitions of $H$ and $\Gamma$, the generalized eigenvalues always exist but they may not converge very rapidly for some problems. The intuition in line 100 is in fact formalized in the estimate (15). The quantity $\nabla \log \pi(w)$ in Algorithm 1 is defined in (28) and (31). We will address other minor questions. We hope that our clarifications can help you for better understanding and reevaluating our contribution.

**To Reviewer 2:** We appreciate your evaluation of the **novelty, high significance, and very sound and careful claims** of our paper, and especially for **your willingness to upgrade the score to full accept after improving the clarity** in Section 3 with the following revision based on your helpful suggestions. Specifically, we will add 1-2 sentences of the intuition for the gradient information matrix for $H$ (the often used gradient-based parameter sensitivity), add 1-2 sentences of the implication by Theorem 1 (for computing the gradient of the projected posterior), and simplify the notations (by slight abuse of notation). We actually have the comparison of cost, see in the caption of Figure 2. A strong motivation for our work is that pSVN uses Hessian matrix which is often not available or too expensive to compute for complex problems. Our examples in Section 4.3 and appendix B are actually the same as those in pSVN, which can be directly compared by examining the two papers on their accuracy and convergence. We will add a short discussion on the comparison of their computational cost in the supplementary material. For more discussion on this see below in To Reviewer 3. Thank you for appreciating the COVID example. We will add a short discussion of it in the main body.

**To Reviewer 3:** Many thanks for your overall very comprehensive and encouraging evaluation of our paper as **sufficiently significant to warrant publication** for our methodological contribution with both theoretical and empirical justifications. On your question about the comparison of pSVGD and pSVN, please see our response above in To Reviewer 2. Moreover, we emphasize that the gradient information matrix in pSVGD only requires the gradients of the log-posterior at the pushed samples, which are already computed in SVGD, see (32). This is often the dominant cost for complex models, e.g., all the examples in this work except for the logistic regression. On the other hand, computing the Hessian in pSVN is much more demanding — (1) evaluating the full Hessian is $d$ times more expensive than evaluating the gradient for $d$-dimensional parameters, (2) evaluating a Gaussian–Newton approximation of the Hessian requires Jacobian, which is $s$ times more expensive for $s$-dimensional data, (3) a low-rank approximation with rank $r$ of the Hessian is $O(r)$ times more expensive by efficient randomized algorithm. We remark that the intrinsic low-dimensional subspace is demonstrated in pSVN for a Bayesian deep learning test problem, which suggests pSVGD is also suitable for such problems, besides our five testing problems from various fields. We will address other minor issues. Thanks again for your very careful reading and **willingness to change the score** given these clarifications.

**To Reviewer 4:** We appreciate your evaluation that our work is **very relevant for Bayesian inference in high dimensions** and our ideas are **novel and give a new approximate inference algorithm**, and that you find the **technical presentation clear and easy to follow.** The convergence guarantee for the gradient information matrix (10) is practically not possible because we can not a-priori draw samples from the posterior but only use samples pushed from the prior to posterior, whose convergence (to the posterior) analysis is beyond the current work. The complexity/scalability in # data points is actually shown in the middle of Figure 4 for pSVGD. Thank you for the suggestion to study the scalability w.r.t. the data dimension, which may lead to development of a new algorithm of projection in data dimension based on the low rankness of the data correlation matrix, which will be further explored. We will address other minor issues.

[Meta-Review · NeurIPS 2020]

Leveraging low-dimensional structure in approximate inference algorithms is an interesting area of study, and this adaptation of SVGD is a promising approach. There were concerns about lack of clarity and presentation of the algorithm, as well as theoretical justification and motivation of this procedure.